# Eating Perception, Nutrition Knowledge and Body Image among Para-Athletes: Practical Challenges in Nutritional Support

**DOI:** 10.3390/nu13093120

**Published:** 2021-09-06

**Authors:** Miwako Deguchi, Hisayo Yokoyama, Nobuko Hongu, Hitoshi Watanabe, Akira Ogita, Daiki Imai, Yuta Suzuki, Kazunobu Okazaki

**Affiliations:** 1Department of Environmental Physiology for Exercise, Osaka City University Graduate School of Medicine, 3-3-138 Sugimoto, Sumiyoshi-ku, Osaka-shi, Osaka 558-8585, Japan; deguchi@osaka-cu.ac.jp (M.D.); dimai@sports.osaka-cu.ac.jp (D.I.); suzuki@sports.osaka-cu.ac.jp (Y.S.); okazaki@sports.osaka-cu.ac.jp (K.O.); 2Department of Food and Human Health Sciences, Osaka City University Graduate School of Human Life Science, 3-3-138 Sugimoto, Sumiyoshi-ku, Osaka-shi, Osaka 558-8585, Japan; kay.hongu@gmail.com; 3Research Center for Urban Health and Sports, Osaka City University, 3-3-138 Sugimoto, Sumiyoshi-ku, Osaka-shi, Osaka 558-8585, Japan; hwatanabe@osaka-cu.ac.jp (H.W.); ogita@osaka-cu.ac.jp (A.O.)

**Keywords:** adult athletes, nutrition education, sports nutrition, dietary practices, online survey, dietitians

## Abstract

Limited information exists on dietary practices in para-athletes. The aim of this study was to clarify the actual situation of para-athletes’ dietary practice and to sort out the factors (i.e., eating perception, nutrition knowledge, and body image), that may hinder their dietary practices, and explored the practical challenges in nutritional support and improving nutrition knowledge for para-athletes. Thirty-two Japanese para-athletes (22 men) and 45 collegiate student athletes without disabilities (27 men) participated in the online survey. The questionnaire included demographic characteristics, eating perception, dietary practices, and nutrition knowledge. The Japanese version of the body appreciation scale was used to determine their body image. Para-athletes who answered that they knew their ideal amount and way of eating showed significantly higher body image scores (r = 0.604, *p* < 0.001). However, mean score for nutrition knowledge of para-athletes were significantly lower than collegiate student athletes (19.4 ± 6.8 vs. 24.2 ± 6.1 points, *p* = 0.001). Both groups did not identify a dietitian as the source of nutrition information or receiving their nutrition advice. The results indicate para-athletes have unique eating perceptions and inadequate nutrition knowledge. Future interventions are needed to examine nutritional supports and education in relation to the role of dietitians.

## 1. Introduction

As physical activity provides health benefits for everyone, a global push is underway to increase physical activity and the participation in sports for people with and without disabilities [1]. Especially for people with disabilities, sports have implications as an effective means for rehabilitation and social inclusion [2,3]. In recent decades, sports for people with disabilities have transformed into competitive sports [4,5,6]. The Paralympic Games is an international event of sporting excellence for athletes with a range of disabilities, which includes mobility disabilities such as amputees, spinal cord injury and cerebral palsy, visual impairments and intellectual disabilities [7,8]. In 2021, 22 athletic events were scheduled in the Paralympic Games, which were held on 24 August to 5 September in Tokyo, Japan (https://www.paralympic.org/tokyo-2020) [9].

Nutrition plays an imperative role in athletic performance [10,11,12]. Adequate macronutrients [13] and micronutrients [14] support intense training as well as optimize physiological function at all competitive levels [15]. Proactive nutrition support by sports dietitians can optimize training outcomes, namely ideal body composition, functional and metabolic adaptations of body, and the performance in the competitions [10,15,16,17]. These nutrition plans need to be personalized to the individual athlete with or without disabilities depending on sports or playing characteristics, and to make performance peak in targeted competitions [18]. On these backgrounds, it has become more common for highly competitive athletes to receive advice from sports dietitians for their daily training [16,18], and further, bring sports dietitians and nutrition staff to international competitions such as the Olympic Games [19]. The practice of sports nutrition for athletes is not only the key to the best performance in competitions, but also a building block of conditioning such as early recovery from fatigue and of prevention of sports injuries and poor physical condition. However, not all athletes, including collegiate student athletes and para-athletes at competitive levels, can receive the support by full-time sports dietitians [20,21] because their university or institution is not able to afford hiring them. In that sense, general collegiate student athletes and para-athletes groups may have the same needs and demand of having sports dietitians beyond their physical conditions or disabilities, age differences, and education levels. It has been reported that collegiate student athletes are not getting benefits from sports nutrition because of their lack of contact with dietitians and cost issues [22]. It is also known that collegiate students who are in the transition to adulthood and are just beginning to become independent of their parents have a peculiar misconception about nutrition [21,22,23]. Evidence suggests there is needs of sports dietitians for competitive athletes. In other words, this is a huge opportunity for dietitians who are interested in working with athletes in various programs at schools or institutions at professional settings. In the United States, the number of sports dietitians in the collegiate setting has grown exponentially, and the efficacy of sports dietitians for collegiate student athletes and para-athletes have been reported [24,25,26].

Para-athletes potentially have various dietary challenges, which depends on individual characteristics of disability [27,28,29]. There actually are very few reports available on dietary intake patterns of para-athletes [12]. The reasons for the lack of research on dietary practices of para-athletes are (1) no general nutritional guidelines for para-athletes exist; (2) para-athletes are not easily grouped by body compositions using standard evaluation tools, such as BMI; (3) physical conditions with several impairment types (i.e., spinal cord injury, upper or lower body amputees, cerebral palsy, etc.) affect the evaluation of energy expenditure [30,31,32] and (4) assistive equipment used by para-athletes may affect lifestyle of para-athletes. However, researchers in various countries in the world are making significant efforts to identify the needs of para-athletes. Some studies reported suboptimal dietary intakes in para-athletes, which may result in low iron and vitamin D [27,30,32,33]. A study of Turkish women wheelchair basketball players (mean age; 25.5 ± 7.2 years) reported inadequate levels of energy consumption, but high a percentage supplied by fat (~44%), and consumption of vitamin B_1_, folic acid, magnesium, iron, fiber, and liquid was insufficient. The study also reported the number of athletes giving correct answers to nutrition knowledge questions were low. The study suggested nutritional counselling/support from a dietitian for high-performance women athletes [34]. Diet quality in Brazilian Paralympic sprinters (male and female athletes, age range; 18~38 years old) was examined using a dietary quality index, the Healthy Eating Index Revised score for the Brazilian population [35]. The study found that Brazilian Paralympic sprinters did not meet the recommendations proposed as a healthy diet, especially with regard to the low intake of milk and dairy products, cereals (especially whole grain), fruits, and vegetables and they had a high intake of solid fat and sugar [36]. Para-athletes often feel inconvenienced in procuring and cooking themselves as they have improper cooking equipment and kitchen environment. Therefore, they may choose highly processed, energy-dense, ready-to-eat food items [12]. These are similar dietary practice patterns of young collegiate athletes who start living independently for the first time with limited cooking skills [37]. In addition, for the para-athletes the amount of food consumed may be restricted to avoid problems with excretion difficulties. Accumulation of evidence about eating behaviors that are unique in para-athletes will help to form the basis of nutrition education by a dietitian. Given the dearth of data on eating perception and nutrition knowledge among para-athletes, the aim of the present study was to examine eating perception—i.e., self-efficacy to make healthy choices and to improve dietary intake, dietary practices, and nutrition knowledge of para-athletes participating in top-level training and collegiate student athletes. Anthropometric and body image scores were also included in order to provide a thorough picture of why para-athletes may not meet the dietary recommendation and propose the need for sport nutrition education by a dietitian to support para-athletes.

## 2. Materials and Methods

### 2.1. Participants

Thirty-two Japanese adult para-athletes with physical disabilities, such as spinal cord injury or limb defect/amputation (see Table 1), participated in the online survey. We recruited the study participants from about 200 athletes with physical disabilities and no other disabilities (such as visual impairment, hearing impairment, and intellectual disabilities or internal diseases) who are engaged in activities such as regular practice at the Osaka Prefectural Exchange Promotion Center for Persons with Disabilities, also known as Fine Plaza Osaka (Sakai City, Osaka, Japan, http://www.fineplaza.jp/ [38]) after approaching staff from the Osaka Para Sports Association (Sakai City, Osaka, Japan, http://www.osad.jp/ [39]). The staff called for study participants from teams that are clearly composed of physically disabled people from the characteristics of the competitions (including wheelchair table tennis, sitting volleyball, etc.) via face-to-face conversations or using leaflets, and recruited para-athletes for the online survey. All the responses obtained were used for the analysis. Neither we nor the staff could identify individuals who responded and answered the questionnaire. Forty-five adult collegiate student athletes without disabilities were also recruited from the members who regularly participated in extracurricular sports club activities in Osaka City University (https://www.osaka-cu.ac.jp/en [40]) and Osaka Prefecture University (https://www.osakafu-u.ac.jp/en/ [41]). To calculate the sample size for the detection of the difference in nutritional knowledge between two groups, we used G*Power software by considering a large effect size (d = 0.8), the α-error (0.05), and the power of (1 − β = 0.80). Based on these assumptions, the desired sample size for each group (para-athletes versus collegiate student athletes) was 26. The institutional Review Board of Osaka City University Graduate School of Medicine approved the study protocol (approval no. 2020-189, approved on 30 September 2020). Informed consent was obtained from all the participants by checking the item showing ‘I agree’ to participate in the online survey. This study also conformed to the ethical guidelines of the 1975 Declaration of Helsinki.

### 2.2. Procedure

The survey was conducted from November 2020 to March 2021. All participants were asked to answer an online survey administered via Questant (Macromill, Inc., Tokyo, Japan, https://questant.jp [42]). Participants accessed an online survey by entering the URL or scanning the Quick Response (QR) code with their smartphone. We also sent a paper questionnaire to those who wished. The questionnaire consisted of five sections: general demographic characteristics (including questions about disabilities), eating behavior, nutrition knowledge, body image, and dietary practice. Participants were allowed to skip any questions they do not want to answer. The time needed to answer the questionnaire was assumed to be 30 min (Appendix A: Survey Questions).

### 2.3. Eating Behavior Questionnaires

The questionnaire about eating behavior is presented in Table 2. The questionnaire consisted of nine eating behavior items: eating for health (eating perception), eating proper amounts and balanced meals, measuring their body weight, satiety responsiveness, and eating favorite and least favorite foods. We used 6 out of 33 items regarding eating for health (eating perception), eating proper amounts and balanced meals, measuring their body weight, satiety responsiveness, and eating favorite and least favorite foods in previously reported questionnaire [43] as a reference when developing the current questionnaire. We revised the questionnaire to find whether they are conscious about their daily diet and trying to eat more healthily or not. Question numbers 2, 4, and 9 were uniquely developed for the following reasons based on our observational experiences working with para-athletes: there are few opportunities for para-athletes to measure their body weight, many para-athletes feel inconvenienced in daily wheelchair operation and transfer, and meals may be eaten regardless of their food preference, in cases where a caregiver prepares their meals. In addition, we focused on the questionnaire of weight maintenance behaviors that para-athletes may avoid excess weight gain. It is interesting to note that the reason for para-athletes to be conscious about weight maintenance, and not gaining weight, is not simply the desire to be thin, but fear of inconvenience in daily wheelchair operations and transfer [44]. Although, low energy availability is also a major problem for both male and female para-athletes [29,30]. All items were rated on a five-point scale (i.e., 1 = never, 2 = seldom, 3 = sometimes, 4 = often, 5 = always). The total score was calculated, ranging from 9 to 45 points. Higher scores indicated healthier eating behavior. To determine the internal consistency, the Cronbach’s alpha was calculated. Cronbach’s alpha values range from 0 to 1, and the minimum requirement for internal consistency has been recommended as 0.7 [45]. The Cronbach’s alpha of this questionnaire was 0.7140.

### 2.4. Nutrition Knowledge Questionnaires

The questionnaire about nutrition knowledge was developed based on two previously reported questionnaires [46,47]. These existing questionnaires were developed to allow athletes from a range of sports and competition levels to rate their nutrition knowledge of current consensus recommendation. The final questionnaires were completed after many stages of verification regarding accuracy, construct validity, internal reliability, and reproducibility [46,47]. We translated these original questionnaires into Japanese and used them, except for some questions that were not relevant to our participants. Furthermore, questions about ingredients that are not commonly consumed in Japan were also excluded. The resulting self-administered questionnaire consisted of 45 items, covering two sections: general nutrition (33 items) and sport nutrition (12 items). These answers were converted to 1 or 0 for correct or incorrect answers, respectively, and were used to calculate the total and subsection scores of nutrition knowledge. The Cronbach’s alpha of this general nutrition and sport nutrition sections were 0.863 and 0.504, respectively. This indicates that internal consistency of the sport nutrition questionnaire was moderate and caution is needed when interpreting the results (see Appendix A: General Nutrition Knowledge and Sport Nutrition Knowledge).

### 2.5. Body Image

To examine body image of the participants, we used the Japanese version of the Body Appreciation Scale-2 (BAS-2) [48]. Most studies exploring body image have focused on negative aspects such as low self-esteem, and pressure of thinness [49]. Within this context, the BAS was developed as a measure reflecting an aspect of positive body image [50], and was updated referred to as the BAS-2, a 10-item measure [51]. Tylka et al., reported that the higher the body appreciation evaluated by BAS-2, the higher the self-esteem, tendency to take adaptive eating behavior, and lower the tendency of physical dissatisfaction and abnormal eating behavior [51]. Items were rated along a five-point scale in the same manner as those in the questionnaire about eating behavior (see Section 2.3). The total score was calculated, ranging from 10 to 50 points. Higher scores indicated a better body image.

### 2.6. Dietary Practice Questionnaires

We set up several questions to sort out the factors that may hinder dietary practices in para-athletes and collegiate student athletes. The questionnaire consisted in the subjective assessment of the amount of daily food and fluid intake, knowledge on the foods needed for improving competitive performance, the source of information about nutrition, and whether they have received support from family members in healthy eating habits. Furthermore, we set up three questions for participants to freely describe: (1) the reason why they answered ‘Adequate’, ‘Not adequate’ or ‘Not sure’ to the question, ‘What do you think about the amount of food and drink you take every day?’; (2) the reason why they answered ‘Not sure (want to know)’ to the question, ‘Do you know what foods you should consume to improve your competitive performance?’ and the way they think they can get the information about the foods they needed; and (3) what they would like to ask dietitians, if anything.

### 2.7. Analysis of Text Data

Among these open-ended responses, text data obtained from participants were quantified by text mining method using a language analysis software, Text Mining Studio, version 6.4 (NTT DATA Mathematical Systems, Tokyo, Japan). First, Japanese text data were divided into grammatically meaningful component units by morphological analysis, then the word relationship network was created to visualize the characteristic words appearing frequently with reference to the participant’s response (i.e., ‘Adequate’, ‘Not adequate’ or ‘Not sure’). Furthermore, the original text written by participants were referred to see how those characteristic words were used in the context [52].

### 2.8. Statistical Analysis

Spearman’s rank correlation coefficient test examined the relationship between the total score of body image and eating behavior. Mann-Whitney U tests were used to compare the score of nutrition knowledge between para-athletes and collegiate student athletes. The chi-squared test was used to compare the differences in answers to the questions about dietary practices between para-athletes and collegiate student athletes. Data were analyzed using SPSS (version 27.0, IBM, New York, NY, USA). P values less than 0.05 were considered statistically significant.

## 3. Results

### 3.1. Characteristics of the Participants

Table 1 shows the physical characteristics of para-athletes. The majority of the para-athletes had spinal or cervical cord injury as their primary disabilities (31.3%, *n* = 10), followed in order by limb defect or amputation (21.9%, *n* = 7) and cerebral palsy (9.4%, *n* = 3). Para-athletes with spinal or cervical cord injury had an injury level of C6-C7 (*n* = 4), T11-T12 (*n* = 2), or L-1 (*n* = 1), and three of them did not answer. More than 85% of the para-athletes used a prosthesis (artificial limbs), a wheelchair, or canes in their daily life. About 40% of the para-athletes had their caregivers prepare their meals (Table 1). Table 3 shows the characteristics of participants. The mean ± standard deviation (SD) age was 40.5 ± 16.3 (ranged from 17 to 72) years in the para-athletes and 21.2 ± 1.0 (ranged from 19 to 23) years in the collegiate student athletes. In both groups, more than 75% of the participants lived with family or a partner. The majority of para-athletes participated in soccer and table tennis as a main sport (both sports—21.9%, *n* = 7), followed by basketball and tennis (both sports—15.6%, *n* = 5). On the other hand, the two major sports of collegiate student athletes were table tennis and baseball (both sports—20.0%, *n* = 9), followed by lacrosse (15.6%, *n* = 7). Thirty-eight percent of para-athletes had more than 20 years of experience playing in the sports in contrast to collegiate student athletes, half of whom had only 1 year or more and less than 5 years of experience.

### 3.2. Relationship between Body Image and Eating Behavior

The total body image and eating behavior scores were 30.8 ± 7.8 and 26.0 ± 7.4, respectively, in para-athletes and 33.8 ± 9.2 and 29.0 ± 5.1, respectively, in collegiate student athletes. There were no statistically significant differences between groups for body image and eating behavior scores (*p* > 0.05). We examined the relationship between body image score and eating behavior score. As shown in Figure 1a, a significant positive correlation was found between the total score of body image and that of eating behavior in para-athletes (r = 0.604, *p* < 0.001). There was no significant correlation between them among collegiate student athletes (r = 0.245, *p* = 0.105) (Figure 1b).

### 3.3. Nutrition Knowledge

Table 4 shows the total and subsection (i.e., general and sport nutrition) scores of nutrition knowledge in both groups. The mean scores for nutrition knowledge of para-athletes were significantly lower than that of collegiate student athletes in both total and sub-section scores (*p* < 0.05). The questions that were answered correctly by 70–90% of the para-athletes were related to basic nutrition knowledge such as questions about whether macronutrients in foods are high or low, and which foods are rich in micronutrients such as vitamin C, iron, and calcium. On the other hand, the rate para-athletes correctly answered was relatively low on specific nutrition knowledge for everyday life, such as the question about caloric contents in 1 g of each of the macronutrients and alcohol (Q27 General Nutrition Knowledge Question), and the percentage of carbohydrate needed for glycogen loading (carbohydrate loading) (Q43, Sport Nutrition Knowledge Question). In para-athletes, mean correct answer rates for these two questions, Q27 and Q43, were 8.6% and 3.1%, respectively. Nearly 80% of para-athletes answered ‘Not sure’ for these questions.

### 3.4. Current Status of Dietary Practices

Table 5 shows the current status of dietary practices in para-athletes and collegiate student athletes. Despite lower nutrition knowledge in para-athletes, the results of subjective assessment of the amount of daily food intake were not significantly different between para-athletes and collegiate student athletes (*p* = 0.120). The same was true for the answers regarding foods needed to consume for improving competitive performance. Meanwhile, the percentage of para-athletes who answered that the amount of their daily fluid intake was ‘Not adequate’ or ‘Not sure’ was significantly higher than that of collegiate student athletes (65.6% vs. 37.8%).

We examined the reasons why participants answered the amount of daily food intake was ‘Adequate’, ‘Not adequate’ or ‘Not sure’ by using text data analysis (see Section 2.7 Analysis of Text Data). Figure 2 shows the word relationship network which visualized the characteristic words appearing frequently in an open-ended response by para-athletes with reference to the subjective assessment of the amount of daily food intake. It was found that the frequent words in free description written by para-athletes who answered ‘Adequate’ were ‘physical condition’, ‘no + problem’ and ‘no + appearance’. Referring to the original text written by para-athletes, these words appeared in the sentences of ‘(my) physical condition is good’, ‘(I have) no problem’ and ‘there is no change in (my) physical condition’. On the other hand, the most frequent words in an open-ended response by collegiate student athletes who answered ‘Adequate’ was ‘(body) weight’. Referring to the original text written by them, the word appeared in the sentences of ‘I have not gained or lost weight’, ‘I can maintain (my) weight’ and ‘(my) weight is appropriate’.

Figure 3 shows the source of the information about nutrition for para-athletes and collegiate student athletes. The largest percentage of participants in both groups answered that the source of the information about nutrition was internet, followed by television. Only one para-athlete identified a dietitian as a source of information about nutrition. Furthermore, 18.8% of para-athletes answered ‘Yes’ to the question, ‘Do you have anything to ask dietitians if you have a chance to talk with them?’ in contrast to 44.4% of collegiate student athletes who answered ‘Yes’. This difference is statistically significant (Table 5). Questions para-athletes wanted to ask a dietitian were more about general nutrition (i.e., advice on future meal preparations, foods that could help their weight loss, their daily total calories,) rather than their sport specific nutrition questions. On the other hand, collegiate student athletes wanted more specific information about sport nutrition from dietitians (i.e., diet to bulk up muscle, sport-event-specific nutrients).

## 4. Discussion

The aim of the present study was to examine eating perception and nutrition knowledge of para-athletes participating in top-level training, currently competing at a national or international level, and collegiate student athletes. This study uniquely tried to understand the para-athletes’ dietary practices, and challenge and opportunities that dietitians may face in implementing nutritional advice in regard to dietary habits for optimizing performance outcomes and overall health among para-athletes. Our main findings were that para-athletes who answered they were practicing the ideal way of eating showed significantly higher body image scores (Figure 1a), while the mean score for nutrition knowledge of para-athletes was significantly lower than that of collegiate student athletes (Table 4).

Body image is affected by various factors such as thoughts, feelings, and behaviors directed towards one’s own body [53]. For instance, negative self-esteem is associated with body dissatisfaction [54]. Paxton et al., evaluated the association between negative self-esteem and body dissatisfaction in a longitudinal study and concluded that body dissatisfaction is a risk factor for increased low levels of self-esteem and depression in adolescent girls and boys [55]. Body image dissatisfaction is one of the symptoms of eating disorders such as anorexia nervosa [56,57], and it has been reported that athletes who participated in aesthetic or weight class sports had high incidence of body image dissatisfaction [58], although sport participation generally improves body satisfaction [59,60]. Comprehensive nutritional studies have reported that body image dissatisfaction may be one of profound risk factors for the development of eating disorders [61,62,63]. However, limited numbers of studies have carried out to examine the relationship between body image dissatisfaction and eating behaviors [64], particularly there were no studies found on athletes taking part in the Paralympic Games. In the present study, a significant positive relationship between the total score of body image and eating behaviors in para-athletes was found. Disability has a negative influence on the psychological experiences, feelings and attitudes towards an individual’s own body [65]. Macdougall et al., reported that para-athletes had lower levels of self-acceptance and body image compared to Olympic sport athletes [66]. Even with such a background, some para-athletes with a positive body esteem might have had better eating behaviors through the improvement of psychological well-being [67] which is one of the key elements of healthier dietary behaviors [68]. Further study should examine what are the unique characteristics of para-athletes who have high body image satisfaction.

Nutrition knowledge affects athletes’ perception of appropriate eating and healthy eating habits [69,70,71]. However, in the present study, the mean scores for nutrition knowledge of para-athletes were significantly lower in both total and in subsections of the questionnaires than those of collegiate student athletes (Table 4). The questionnaire about general nutrition knowledge used in the current study consisted of four categories: macronutrient, micronutrient, fluid, and alcohol. Para-athletes had lower scores in general nutrition knowledge except for questions about fluid intake compared to collegiate student athletes. In the present study, most para-athletes were unaware of the types of fats they should reduce in their diet (Q21 General Nutrition Knowledge Question), despite their attention to weight gain and lifestyle-related diseases such as hypertension and diabetes because of high susceptibility of people with spinal cord injury to those diseases [72]. Furthermore, only 3.1% of para-athletes correctly answered the question about the percentage of dietary carbohydrate needed for glycogen (carbohydrate) loading. Glycogen (carbohydrate) loading is a basic knowledge in sports conditioning as a means of increasing intramuscular glycogen stores [73], and improve performance [74,75]. Thus, even para-athletes at a high competition level do not have basic knowledge of sports nutrition, and conversely, there is room for improvement in competitive performance through the transition of sports nutrition knowledge to them.

There is limited study on nutritional needs for para-athletes [12]. The lack of established sports nutrition strategy for para-athletes may result in fewer opportunities for nutrition education and support by dietitians. In fact, para-athletes receiving the Brazilian Federal sport scholarship program were no more likely to have nutritional support compared to nonscholarship para-athletes [76]. Nutrition education is obviously one of the methods to improve nutrition knowledge. Rastmanesh et al., reported that nutrition education for athletes and their coaches with spinal cord injury or amputation not only improved their nutrition knowledge score and increased mean micronutrients and fiber intake, but also increased awareness of opportunities to receive nutrition information and supports from dietitians [25]. The study suggested that para-athletes may benefit from more nutritional education and counseling offered by dietitians in collaboration with their coaches to improve health and athletic performance. Another study also reported that nutritional advice could improve their knowledge of the benefits of sports nutrition [26]. Although it is needed to verify whether the improvement of nutrition knowledge is linked to better eating behaviors, the evidence of nutrition education for para-athletes is accumulating in this way.

Despite lower nutrition knowledge in the para-athletes, about 40% of them answered that they thought the amount of daily food they took was adequate, and that they knew certain foods improved competition performance. However, their mastery feelings seem to be based not their nutrition knowledge, but on their own subjective indicator. In other words, they may have determined that they were eating properly because they were in good physical condition (Figure 2). This was in contrast to collegiate student athletes assessing the amount of daily food intake based on their body weight, an objective indicator. There are a few possible reasons why para-athletes rely on their own subjective indicator. First, there are few opportunities for para-athletes to measure body weight, the simplest and most reliable way to assess energy balance. It is difficult for most people with lower-limb disability to keep a standing position on the scale and wheelchair-accessible weighing scales are not available for everyone. In fact, in the present study, 53.1% of para-athletes answered they ‘never’ or ‘seldom’ measure body weight. Second is the difficulty in estimating the resting metabolic rate from the characteristics of their body composition. Skeletal muscle atrophy, especially the muscles of the lower extremities, is the most common feature of spinal cord injury. People with spinal cord injury had smaller skeletal muscle cross-sectional area compared to the abled-bodied population [77]. Furthermore, muscle atrophy and reduction in muscle strength are also observed in lower limb amputees [78,79]. These characteristics make it difficult to estimate their resting metabolic rate using the existing estimation formula for adults without disabilities. Future research is needed for proactive nutrition support to para-athletes to establish a method that estimate their resting metabolic rate and daily energy requirement.

With respect to fluid intake, the correct answer rate for the question regarding the daily fluid requirement in para-athletes was slightly higher than that in collegiate student athletes, although the difference was not statistically significant (62.5% vs. 40.0%, *p* = 0.052). Nevertheless, about 65% of para-athletes answered that the amount of daily fluid they took was ‘Not adequate’ or that they were ‘Not sure’. These results suggest that para-athletes cannot meet fluid requirements even if they know the amount of fluid needed to consume, or they are not even aware of how much fluid to take. Patients with spinal or cervical cord injury are faced with autonomic dysfunctions and with the inability to lose excess heat by sweating [80,81]. Therefore, hydration management for para-athletes is important to prevent hyperthermia and deterioration in competitive performance. However, para-athletes tend to restrict fluid intake due to the practical problem in lavatory access and urination management. Hydration strategies for para-athletes become increasingly important due to their unique thermoregulation and hydration challenges [82,83]. Further analyses are needed to deepen understanding of individual situations regarding fluid replacement needs and hydration difficulties in para-athletes.

Competitive athletes without disabilities not only receive advice and support from dietitians/nutritionists for their daily training but also bring them as staff to international competitions such as the Olympic Games [19]. However, proactive nutritional support for para-athletes lags far behind. It was reported that only 18% of wheelchair basketball female athletes identified a dietitian as their source of nutrition knowledge [34]. Our results in which few para-athletes received information from a dietitian were consistent with the previous report. Furthermore, the percentage of participants who answered that they had something to ask a dietitian was significantly lower in para-athletes than in collegiate student athletes. Questions in case they had something to ask a dietitian were vague and lacking in specificity (e.g., “What is a good food for me?” “Which foods makes me stronger?” “What is fast food?”). These results suggested that para-athletes may not know what they could ask a dietitian, such as how to improve their physical health and performance, prevent sports related injuries, or reduce fatigue, and individualized nutritional advice. Therefore, it is important to develop support systems that may promote involving para-athletes with dietitians/nutritionists. Although, we do not have the data of their educational background, to develop a good partnership with a dietitian for para-athletes, educational background may be checked in a future intervention study.

A few limitations of the present study should be noted. First, the numbers of participants in para-athletes were small. These might have been insufficient to sort out the factor that may hinder practice of sport nutrition for para-athletes. Second, we developed current nutrition knowledge questionnaire by translating the existing nutrition knowledge questionnaires in Japanese, followed by selecting and adjusted some wording without changing original contents. Prior to the study we have not verified the reliability of current questionnaire as to whether nutritional knowledge in our participants could be evaluated. The Cronbach’s alpha of the sports nutrition section of current questionnaire also did not achieve an adequate value for internal consistency. Therefore, in a future study the questionnaire needs to be improved to be highly reliable to evaluate nutrition knowledge. Furthermore, even though the questionnaires about nutrition knowledge used as references in the current study have been fully validated, not all the correct answers set for each question are necessarily relevant to our study participants. That is, the reliability of the questionnaire also depends on the attributes (especially in para-athletes) and nationality of the participant because the factors affect consensus recommendation. Therefore, to promote nutritional support for para-athletes, a questionnaire to evaluate their nutritional knowledge must be developed and validated for para-athletes in the future. Third, as we mentioned earlier, we did not obtain the information about the educational background of para-athletes. Educational background may have contributed their lower nutrition knowledge compared to collegiate student athletes, which may have affected the interpretation of the results. Finally, in the present study, actual dietary intake of participants was not examined. Therefore, whether para-athletes met the required amount of nutrients, or they consumed the food needed for improving competitive performance was unknown. It is known that people overestimate vegetable intake, well-known healthy foods, than how much they actually took [84]. Thus, we need to have follow-up studies assessing the actual dietary intake and eating perceptions of para-athletes in order to identify possible motivational nutritional strategies in the future.

## 5. Conclusions

Para-athletes who answered they knew the ideal amount to eat and practiced healthy eating showed significantly higher body image scores. On the other hand, their confidence in having a proper diet was not supported by nutritional knowledge, which could be at least in part due to their lack of objective indicators of energy balance as well as their lack of involvement with dietitians. In a situation where the field of sports nutrition for para-athletes is still developing, there is a need for development of the knowledge transfer partnership between para-athletes and dietitians. Future interventions translating sports nutrition principles to find effective nutrition education methods especially for para-athletes are needed.

## Figures and Tables

**Figure 1 nutrients-13-03120-f001:**
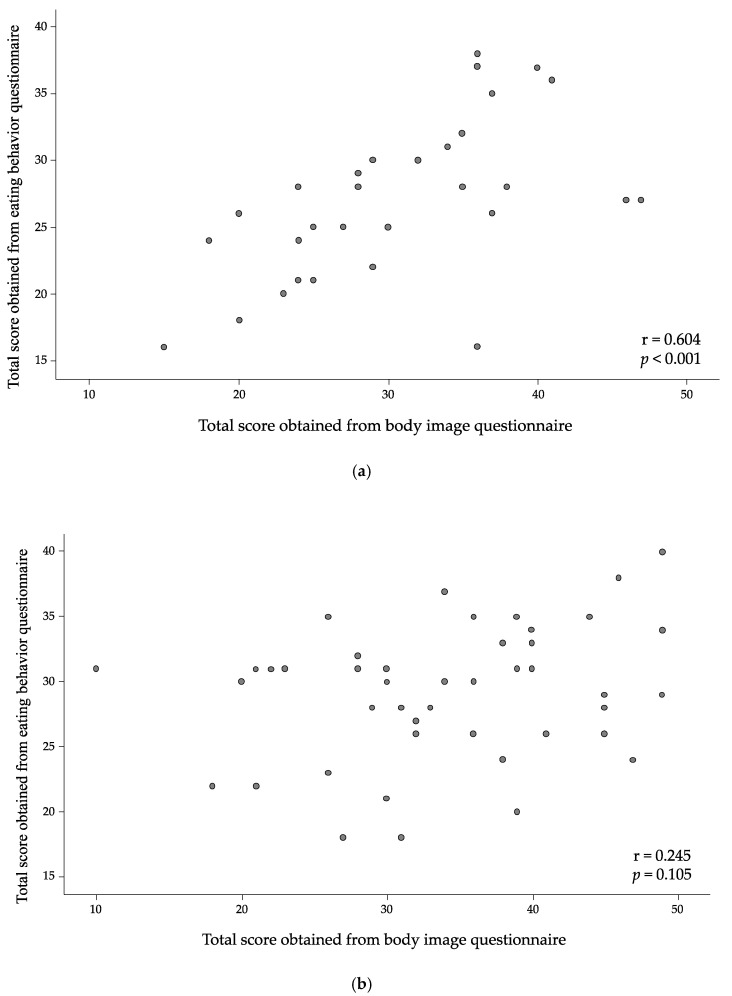
Relationship between body image score and eating behavior score in para-athletes (**a**) and collegiate student athletes (**b**). Total score obtained from body image questionnaire and eating behavior questionnaire showed a significant positive correlation in para-athletes (r = 0.604, *p* < 0.001).

**Figure 2 nutrients-13-03120-f002:**
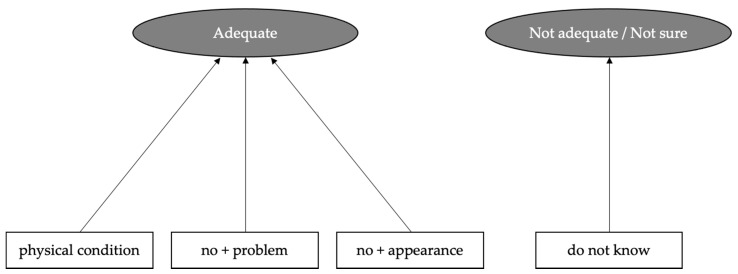
Analysis of word relationships visualizing the characteristic words that appear frequently in an open-ended response by para-athletes with reference to the subjective assessment of the amount of daily food intake (i.e., ‘Adequate’ or ‘Not adequate’/‘Not sure’). Colored circles indicate the answers to the subjective assessment of the amount of daily food intake and white squares indicate words that appear frequently in an open-ended response. The frequent words in an open-ended response by para-athletes who answered ‘Adequate’ were ‘physical condition’, ‘no + problem’ and ‘no + appearance’.

**Figure 3 nutrients-13-03120-f003:**
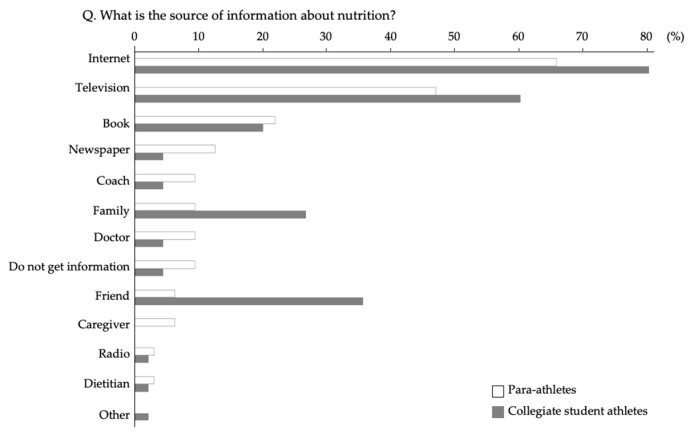
The percentage of participants who answered that each item was applicable to the question ‘*What is the source of information about nutrition?*’ The participants were allowed to select one or more items.

**Table 1 nutrients-13-03120-t001:** Physical characteristics of para-athletes.

		*n*	(%)
Major types of physical disabilities	Spinal or cervical cord injury	10	(31.3)
	Limb defect or amputation	7	(21.9)
	Cerebral palsy	3	(9.4)
	Other	8	(25.0)
	Unanswered	4	(12.5)
Onset of the disabilities	Congenital	11	(34.4)
	Acquired	17	(53.1)
	Unanswered	4	(12.5)
Limb defect/amputation	Yes	8	(25.0)
	No	21	(65.6)
	Unanswered	3	(9.4)
Use of prosthesis (artificial limbs)	Yes	7	(21.9)
	No	22	(68.8)
	Unanswered	3	(9.4)
Use of wheelchair	Yes	23	(71.9)
	No	7	(21.9)
	Unanswered	2	(6.3)
Use of a cane/canes	Yes	6	(18.8)
	No	24	(75.0)
	Unanswered	2	(6.3)
Presence of bedsore	Yes	4	(12.5)
	No	26	(81.3)
	Unanswered	2	(6.3)
Presence of caregiver	Yes	16	(50.0)
	No	15	(46.9)
	Unanswered	1	(3.1)
Contents of supports requested to a caregiver *	Daily shopping	11	(34.4)
	Meal preparation	12	(37.5)
	Dressing	6	(18.8)
	Movement	6	(18.8)
	Excretion	5	(15.6)

* The participants were allowed to select one or more items.

**Table 2 nutrients-13-03120-t002:** Eating behavior questionnaires.

	Questions
1	I am careful about what to eat for my health.
2	I measure my body weight.
3	I eat meals on a regular schedule.
4	I think about my weight (not to gain weight), when I eat meals.
5	I serve well-proportioned foods (enough amount) on my plate.
6	I try to stop eating, even if I want to eat a little more.
7	I sometimes refrain from taking a foods for the sake of my body, even if I like it.
8	I incorporate a food into my diet for the sake of my body, even if I don’t like it.
9	I eat any food served on my plate, even if I dislike it.

**Table 3 nutrients-13-03120-t003:** Characteristics of the participants.

		Para-Athletes		Collegiate Student Athletes	
		*n*	(%)	*n*	(%)
Gender	Men	22	(68.8)	27	(60.0)
	Women	10	(31.3)	18	(40.0)
Age (years)	10–19	2	(6.3)	2	(4.4)
	20–29	8	(25.0)	42	(93.3)
	30–39	6	(18.8)	0	(0.0)
	40–49	6	(18.8)	0	(0.0)
	50–59	5	(15.6)	0	(0.0)
	60–69	2	(6.3)	0	(0.0)
	70–79	3	(9.4)	0	(0.0)
	Unanswered	0	(0.0)	1	(2.2)
	Mean	40.5		21.2	
	SD	16.3		1.0	
Style of living	Living alone	5	(15.6)	10	(22.2)
	Living with family or partner	25	(78.1)	35	(77.8)
	Other	2	(6.3)	0	(0.0)
Main sport played	Table tennis	7	(21.9)	9	(20.0)
	Baseball	0	(0.0)	9	(20.0)
	Lacrosse	0	(0.0)	7	(15.6)
	Soccer	7	(21.9)	0	(0.0)
	Dance	1	(3.1)	6	(13.3)
	Nippon kempo (Japanese martial art)	0	(0.0)	6	(13.3)
	Basketball	5	(15.6)	0	(0.0)
	Tennis	5	(15.6)	0	(0.0)
	Track and field	0	(0.0)	5	(11.1)
	Swimming	4	(12.5)	0	(0.0)
	Japanese archery	0	(0.0)	2	(4.4)
	Volleyball	1	(3.1)	1	(2.2)
Years of competition (years)	<1	0	(0.0)	1	(2.2)
	1–4	4	(12.5)	21	(46.7)
	5–9	5	(15.6)	11	(24.4)
	10–14	6	(18.8)	8	(17.8)
	15–19	3	(9.4)	5	(11.1)
	≧20	12	(37.5)	0	(0.0)

**Table 4 nutrients-13-03120-t004:** The score of nutrition knowledge questionnaire obtained from para-athletes and collegiate student athletes.

		Para-Athletes			Collegiate Student Athletes			*p*-Value
Knowledge Section (Max Score)		Mean	SD	Score Rate (%)	Mean	SD	Score Rate (%)	
Total	(45)	19.4	6.8	43.1	24.2	6.1	53.7	0.001
General nutrition	(33)	15.0	5.5	45.6	18.3	5.1	55.4	0.011
Sport nutrition	(12)	4.3	2.2	36.2	5.9	1.9	48.9	0.004

The data were presented as mean, SD and score rate (%). The scores of total and each section were compared by Mann–Whitney U-test.

**Table 5 nutrients-13-03120-t005:** Dietary practices in para-athletes and collegiate student athletes.

		Para-Athletes		Collegiate Student Athletes		*p*-Value
		*n*	(%)	*n*	(%)	
Subjective assessment of the amount of daily food intake	Adequate	15	(46.9)	30	(66.7)	0.120
	Not adequate	7	(21.9)	10	(22.2)	
	Not sure	9	(28.1)	5	(11.1)	
Subjective assessment of the amount of daily fluid intake	Adequate	10	(31.3)	28	(62.2)	0.020
	Not adequate	14	(43.8)	14	(31.1)	
	Not sure	7	(21.9)	3	(6.7)	
Foods needed to consume for improving competitive performance	Know	3	(9.4)	6	(13.3)	0.250
	Rougly know	11	(34.4)	21	(46.7)	
	Not sure (want to know)	9	(28.1)	14	(31.1)	
	Not sure (do not want to know)	8	(25.0)	4	(8.9)	
Support from family in practicing healthy lifestyle	Supportive	11	(34.4)	6	(13.3)	0.058
	Rather supportive	18	(56.3)	21	(46.7)	
	Rather not supportive	3	(9.4)	14	(31.1)	
	Not supportive	0	(0.0)	4	(8.9)	
Something one wants to ask dietitian	Yes	6	(18.8)	20	(44.4)	0.029
	No	24	(75.0)	25	(55.6)	
	Unanswered	2	(6.3)	0	(0.0)	

The differences in answers to each question about dietary practices between the groups were examined by chi-squared test.

## Data Availability

The data presented in this study are openly available in FigShare at doi:10.6084/m9.figshare.15059394.

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
