# Peer review of "Eating Perception, Nutrition Knowledge and Body Image among Para-Athletes: Practical Challenges in Nutritional Support"

_nutrients, 2021, doi:10.3390/nu13093120_

Round 1

Reviewer 1 Report

This manuscript entitled "Eating Perception, Nutrition Knowledge and Body Image among Para-Athletes: Practical Challenges in Nutritional Support" aimed to larify the actual situation of para-athletes’ dietary practice and to sort out the factors (i.e., eating perception, nutrition knowledge, and body image), that may hinder their dietary practices.

The manuscript is very interesting. However, some issues should be addressed by the authors:

ABSTRACT

  • Include a background sentence.
  • Just include male or female value. The remainder is deducible.
  • Conclusion is not clear enough.
  • Keywords:  do not repeat from the title.

INTRODUCTION

  • It is important for the background to include recent references about this topic. Please, bring results from other countries in this section.
  • Rationale should be improved.

RESULTS

  • Figure 2 should be improved ou removed.
  • Figure 3 should be improved.

REFERENCES

  • Several recent articles from Nutrients could be cited.

Author Response

We greatly appreciate Reviewer 1 for careful peer review and kind comments. We have made corrections to each of the issues. Our responses are as follows:

ABSTRACT

Include a background sentence.

We added the background sentence:

Limited information exists on dietary practices in Para-athletes. (Line 15)

Just include male or female value. The remainder is deductible.

We included only male values in both groups. (Lines 19 and 20)

Conclusion is not clear enough.

Thank you for your suggestion. We have revised the conclusion statement based on the results of this study as follows:

The results indicate para-athletes have unique eating perceptions and inadequate nutrition knowledge. Future interventions are needed to examine nutritional supports and education in relation to the role of the dietitians. (Lines 26-29)

Keywords:  do not repeat from the title.

We removed the same words from the title and added new keywords;
adult athletes; nutrition education; sports nutrition, dietary practices, online survey; dietitians (Lines 30-31)

INTRODUCTION

It is important for the background to include recent references about this topic.

Reference #4 was replaced to a recent reference and we added more recent references.

Removed old Reference #4, Clark, M.W. Competitive sports for the disabled. Am J Sports Med 1980, 8, 366-369, doi:10.1177/036354658000800514.

Replaced Reference #4, Blauwet C, Willick SE. The Paralympic Movement: using sports to promote health, disability rights, and social integration for athletes with disabilities. PM R. 2012 Nov;4(11):851-6. doi: 10.1016/j.pmrj.2012.08.015. PMID: 23174549.

Added Reference #5, Silver JR. The origins of sport for disabled people. J R Coll Physicians Edinb. 2018 Jun;48(2):175-180. doi: 10.4997/JRCPE.2018.213. PMID: 29992210.

Reference # 16, Devlin BL, Leveritt MD, Kingsley M, Belski R. Dietary Intake, Body Composition, and Nutrition Knowledge of Australian Football and Soccer Players: Implications for Sports Nutrition Professionals in Practice. Int J Sport Nutr Exerc Metab. 2017 Apr;27(2):130-138. doi: 10.1123/ijsnem.2016-0191. Epub 2016 Oct 6. PMID: 27710165.

 Please, bring results from other countries in this section.

List of countries in this section:
Reference #16 - Australian Football and Soccer Players
Reference #17 - USA female volleyball team
Reference #18 - athletes village at the Sydney
Reference #19 - USA Division III football players
Reference #20 - Iranian college athletes
Reference #21 – USA division I college athletes
Reference #22 - Athletes from five different private colleges in Salem District, India
Reference #24 - an elite wheelchair marathoner in Spain
Reference #25 - Swiss elite wheelchair athletes
Reference #27 – young Paralympic swimmers in Mexico
Reference #28 – USA and Canadian Male and female participants (international and national level para-athletes (wheelchair basketball and track)
Reference #30 Turkish national women wheelchair basketball players
Reference #32 – Brazilian Track-and -Field Paralympic athletes

Rationale should be improved.

We have revised the rational of this study including questions that was asked by Reviewer 2 (L71–74: Since the most of the background discussion is on para-athletes, I could not read why college athlete students were also examined) and Reviewer 3 (In the introduction, please include a couple of examples of the dietary challenges experienced by para-athletes (lines 65-66)).

RESULTS

Figure 2 should be improved or removed.

We revised Figure 2 and the footnote to make it easier to understand.

Figure 3 should be improved.

We revised Figure 3 and the footnote to make it easier to understand.

REFERENCES

Several recent articles from Nutrients could be cited.

We have added 5 recent articles from Nutrients (References #12, #17, #25, #26, and #28).

Reviewer 2 Report

This paper sought to investigate the comparison between para-athletes and college athlete students on general and sports nutrition knowledge and eating behaviors. There were some questions about the methods such as participants and questionnaires of this study. The following is a summary of some of the points of concern for your reference.

[Major]
L33–34: It seems to me that the health benefits mentioned are physical activity, and sports are one part of it.
L49–51: Literature 17 does not seem to be a study on athletes.
L71–74: Since the most of the background discussion is on para-athletes, I could not read why college athlete students were also examined.
L79: When was the survey conducted? How many people were in the population and how were the participants selected? How did you calculate the study size?
L82-85: What are the reasons for excluding participants with disabling locations or diseases? I could not read that reason from the background.
L95–96: I think it is inconsistent in terms of eligibility criteria to not exclude participants whose major types of disabilities are "other" or "unanswered."
L108–116: There needs to be mention of the validation. What is the difference between item numbers 6 and 7? What is the difference between item numbers 8 and 9?
L117: I could not read the reason for the special focus on the obesity prevention. I think preventing the desire for thinness items are also needed within the question. I think the current scale clearly contradicts the positive score in item number 10 of body image.
L119–120: Please provide more details because it relates to interpretation of validity. For example, the ratio of carbohydrate and the recommended dietary allowance of protein items in reference number 29 seem to be relevant to the participants in this study. Also, please state the reliability of this questionnaire.
L435–790: For Q25, it seems that spinach is also correct in the source. For Q28, according to DRIs, it seems that no reference report has been found in Japan. For Q29, dizziness and nauseas are thought to be 3-6%, what is the source? For Q32, 10g also seems to be in the appropriate range, and 30g may apply to men as well.

[Minor]
L102–104: Where did you include questions about disability in the section?
L188: The words "Men/Women" in the table are different in line 441–442.
L791–936: The first letter of the word is mixed up with upper and lower case, so please check the instructions.

Author Response

We greatly appreciate Reviewer 2 for kind and detailed review. We have made corrections to each of the issues. Our responses are as follows:

L33–34: It seems to me that the health benefits mentioned are physical activity, and sports are one part of it.

Good point.  We have revised the sentence as follows:

 As physical activity provides health benefits for everyone, global push is underway to increase physical activity in whole populations and strongly promotes the participation in sports for people with and without disabilities [1]. (Lines 34-36)

L49–51: Literature 17 does not seem to be a study on athletes  

              We removed literature 17.

L71–74: Since the most of the background discussion is on para-athletes, I could not read why college athlete students were also examined.

In Introduction we have explained why college athlete students were included in this study. (Lines 64-92)

L79: When was the survey conducted? How many people were in the population and how were the participants selected? How did you calculate the study size?

As we mentioned in 2.2 Procedure, the survey was conducted from Nov 2020 to Mar 2021. From about 200 athletes with physical disabilities who are engaged in activities such as regular practice at the Osaka Prefectural Exchange Promotion Center for Persons with Disabilities, also known as Fine Plaza Osaka (Sakai City, Osaka, Japan, http://www.fineplaza.jp/), the Osaka Para Sports Association (Sakai City, Osaka, Japan, http://www.osad.jp/) staff called for participation face-to-face or using leaflets and recruited them for the survey. We also failed to show in the original version how to calculate the sample size. We calculated the sample size for the detection of the difference in nutritional knowledge between the groups using G*Power software by considering a large effect size (d = 0.8), the α-error (0.05), and the power we aimed (1 − β = 0.80). Based on these assumptions, the desired sample size for each group (para-athletes versus collegiate student athletes) was 26.

We added the above content to the revised manuscript. (Lines 106-111, lines 118-122)

L82-85: What are the reasons for excluding participants with disabling locations or diseases? I could not read that reason from the background.

L95–96: I think it is inconsistent in terms of eligibility criteria to not exclude participants whose major types of disabilities are "other" or "unanswered."

Regarding the exclusion criteria that the reviewer 2 mentioned, at first, we thought that it did not meet the purpose of this study, if athletes with defect/amputation limited to the upper limbs were included in this study. The reason was, like other disabilities such as hearing impairment, having a defect/amputation limited to the upper limbs may not be involved significant movement problems and excretion difficulties. Therefore, in the original manuscript we have performed the statistical analysis excluding these participants. Thank you to Reviewer 2, we have checked the conditions of athletes with a defect/amputation limited to the upper limbs and found that they have the difficulties in dietary practice as those with disabilities in the lower limbs due to the inconvenience in procuring ingredients and cooking. Consequently, we re-analyzed all 32 answers from para-athletes including those with defect/amputation limited to the upper limbs and revised the Result section.

The participants who answered their major types of disabilities are "Other" or "Unanswered" remained to be included as in the original version because it was confirmed by the staff of the Osaka Para-Sports Association that they were physically disabled. Accordingly, we modified the Participants section and Table 1.

L108–116: There needs to be mention of the validation. What is the difference between item numbers 6 and 7? What is the difference between item numbers 8 and 9?

The eating behavior questionnaires in this survey consisted of items about eating for health (eating perception), eating proper amounts and balanced meals, measuring body weight, satiety responsiveness, and eating favorite and non-favorite foods. The internal consistency of this questionnaire examined by Cronbach’s alpha was appropriate (= 0.7083). Since no significant correlation was found between the score of item No. 6 and 7 (r = 0.247, p = 0.102) or between the score of item No. 8 and 9 (r = 0.348, p = 0.095), we don’t think there is any duplication in each item. However, as Reviewer 2 pointed out, the English translation of the questionnaire in the original version may not have been able to clearly show the difference between each item. Therefore, we revised Table 2 as follows:

Table 2. Eating behavior questionnaires.

Questions

1

I am careful about what to eat for my health.

2

I measure my body weight.

3

I eat meals on a regular schedule.

4

I think about my weight (not to gain weight), when I eat meals.

5

I serve well-proportioned foods (enough amount) on my plate.

6

I try to stop eating, even if I want to eat a little more.

7

I sometimes refrain from taking a food for the sake of my body, even if I like it.

8

I incorporate a food into my diet for the sake of my body, even if I don’t like it.

9

I eat any food served on my plate, even if I dislike it.

L117: I could not read the reason for the special focus on the obesity prevention. I think preventing the desire for thinness items are also needed within the question. I think the current scale clearly contradicts the positive score in item number 10 of body image.

There is no specific esthetic sport in competition for athletes with physical disabilities. In addition, it has been reported that many para-athletes reduce their food intake because they are afraid of inconvenience in daily wheelchair operation and transfer rather than from a desire for thinness (Broad E. Chapter 6 Spinal Cord Injuries in 2nd Edition of Sports Nutrition for Paralympic Athletes, 2019). That is why we focused on behaviors to avoid weight gain in the eating perception questionnaires. Low energy availability is also a major problem for para-athletes (Egger T, et al. Nutrients 12(11): 3262, 2020; Pritchett K, et al. Nutrients 13(3): 979, 2021). On the other hand, Body Appreciation Scale is not just about how para-athletes perceive their body weight but reflects their respect and self-efficacy for own body. Therefore, we believe that there is no contradiction between these items.

We added the above content to the revised manuscript. (Lines 151-157)

L119–120: Please provide more details because it relates to interpretation of validity. For example, the ratio of carbohydrate and the recommended dietary allowance of protein items in reference number 29 seem to be relevant to the participants in this study. Also, please state the reliability of this questionnaire.

L435–790: For Q25, it seems that spinach is also correct in the source. For Q28, according to DRIs, it seems that no reference report has been found in Japan. For Q29, dizziness and nauseas are thought to be 3-6%, what is the source? For Q32, 10g also seems to be in the appropriate range, and 30g may apply to men as well.

The issue Reviewer 2 mentioned above is a very important point. The questionnaires used in the current study were developed to allow athletes from a range of sports and competition levels to rate their knowledge of current consensus recommendation. The final questionnaires   were completed after many stages of verification regarding accuracy, construct validity, internal reliability, and reproducibility. However, not all the correct answers set for each question are necessarily valid. In addition, the reliability of the questionnaire depends on the attributes (especially in para-athletes) and nationality of the participant because the factors affect consensus recommendation. In addition, as Reviewer 2 pointed out, the Ministry of Health, Labor and Welfare has not established a clear guideline for water required yet, despite they stated in the 2020 version of Dietary Reference Intake citing the report by Sawka MN, et al. (Human water needs. Nutr Rev, 2005) that the amount of water required was estimated to be about 2.3 to 2.5 L/day for the group with low physical activity level and 3.3 to 3.5 L/day for the group with high physical activity level. Therefore, to promote nutritional support for para-athletes, a questionnaire to evaluate their nutritional knowledge must be developed and validated in the future.

We mentioned the issue in the Limitation section in the revised version. (Lines 456-463)

We also added sentences about the questionnaires used in the current study on the Nutrition Knowledge Questionnaires section as follows:

These existing questionnaires were developed to allow athletes from a range of sports and competition levels to rate their knowledge of current consensus recommendation. The final questionnaires were completed after many stages of verification regarding accuracy, construct validity, internal reliability, and reproducibility [37, 38]. (Lines 166-170)

Regarding Q32, the question was about the upper limit of the appropriate amount of alcohol, so the Appendix was revised as follows:

Q32. How much pure alcohol is defined as ‘upper limit of the appropriate alcohol consumption’ by Ministry of Health, Labour, and Welfare (Line 673)

L102–104: Where did you include questions about disability in the section?

The questions about disability were included general demographic characteristics section.

According to Reviewer 2’ suggestion, we revised the sentence as follows:

We also sent a paper questionnaire to those who wish. The questionnaire consisted of five sections: general demographic characteristics (including questions about disabilities), eating behavior, nutrition knowledge, body image, and dietary practice. (Lines 135-137)

L188: The words "Men/Women" in the table are different in line 441–442.

We changed the words “Male/Female” in line 441-442 to “Men/Women”. (Lines 501-502)

L791–936: The first letter of the word is mixed up with upper and lower case, so please check the instructions.

We unified the first letter of a word in References section to lowercase.

Reviewer 3 Report

The manuscript describes the findings of a survey study conducted to assess eating perception, nutritional knowledge, and body image among para-athletes and compare these items between para-athletes and collegiate athletes. The research is novel and provides an important foundation to fill a knowledge gap that exists in the specific population of para-athletes. Overall, the methodology was appropriate for a small-scale exploratory study and is able to provide preliminary findings on which to build a more extensive evidence base in the future. The authors adequately described the limitations of the methodology employed in the discussion section. There are several minor areas that could use improvement, especially to provide more context to readers who are not familiar with the para-athlete population. Please find my specific comments below:

  • In the introduction, please include a couple of examples of the dietary challenges experienced by para-athletes (lines 65-66).
  • Please provide some more information about the eating behavior questionnaire development. How were the items chosen/developed? Were any items taken from previously validated scales?
  • Please clarify why only para-athletes were asked the follow-up questions described from line 147-152.
  • From line 190-191, please clarify which values are which. For example, potentially revise the sentence from “The total score of body image and that of eating behavior in para-athletes and collegiate student athletes were 30.5 vs. 33.8 and 25.8 vs. 29.0, respectively.” to “The total body image and eating behavior scores were 30.5 and 25.8, respectively, in para-athletes and 33.8 and 29.0, respectively, in collegiate athletes.”
  • In Table 5, please modify the spacing so it is clearer which items go with which responses.
  • In all tables that report p-values, please include a note describing which statistical tests were used for easy reference.
  • On line 325, please explain what is meant by lifestyle-related disease and/or provide a couple of examples.
  • On line 351-352, please explain why para-athletes do not have many opportunities to measure their body weight.
  • The sentence from line 368-369 is somewhat confusing because the participants responded that they were either not consuming enough fluid or they were unsure if they were consuming enough. Therefore, it doesn’t seem completely accurate to say that the majority of para-athletes could not meet their fluid requirements despite knowing what they are unless you refer to just the group that answered ‘not adequate’.
  • Finally, the paper could use some editing to ensure proper English throughout, which will increase its readability.

Author Response

We greatly appreciate Reviewer 3 for kind review and favorable comments. We have made corrections to each of the issues. Our responses are as follows:

In the introduction, please include a couple of examples of the dietary challenges experienced by para-athletes (lines 65-66).

Good point.  We have explained about the dietary challenges experienced by para-athletes using the two published papers, i.e., 1) Turkish national women wheelchair basketball players (Reference # 30) and 2) Brazilian Track-and -Field Paralympic male and female athletes (Reference # 32). 

Please provide some more information about the eating behavior questionnaire development. How were the items chosen/developed? Were any items taken from previously validated scales?

We developed the eating behavior questionnaire based partly on previously reported questionnaire (Takayama N, et al. Kenkouigaku 21(1): 28, 2012). We revised the questionnaire to ask if they are eating healthier. Item numbers 2, 4, and 9 were uniquely developed for the following reasons, respectively: there are few opportunities for para-athletes to measure body weight, the simplest and most reliable way to assess energy balance; many para-athletes are afraid of inconvenience in daily wheelchair operation and transfer; in case a caregiver prepares their meals, a food may be served regardless of their preference.

We added the above content to the revised manuscript. (Lines 144-151)

Please clarify why only para-athletes were asked the follow-up questions described from line 147-152.

We actually set up these follow-up questions not only for para-athletes but for collegiate student athletes. Similarly, we also analyzed text data among the open-ended responses for collegiate student athletes. Accordingly, we have revised the sentence as follows:

Furthermore, we set up three questions for participants to freely describe: 1)… (Lines 195-196)

Among these open-ended responses, text data obtained from participants were quantified by text mining method using a language analysis software, Text Mining Studio, version 6.4 (NTT DATA Mathematical Systems, Tokyo, Japan). (Lines 203-205)

We have examined the reasons why participants answered the amount of daily food intake was ‘Adequate’, ‘Not adequate’ or ‘Not sure’ by using text data analysis (See 2.7. Analysis of Text Data). (Lines 286-288)       

We also added sentences about the responses from collegiate student athletes to the follow-up questions on Result and Discussion section as follows:

On the other hand, the most frequent words in an open-ended response by collegiate student athletes who answered ‘Adequate’ was ‘(body) weight’. Referring to the original text written by them, the word appeared in the sentences of ‘I have not gained or lost weight’, ‘I can maintain (my) weight’ and ‘(my) weight is appropriate’. (Lines 295-298)

On the other hand, collegiate student athletes wanted more specific information about sport nutrition from dietitians (i.e., diet to bulk up muscle, sport event-specific necessary nutrients). (Lines 320-322)

This was in contrast to collegiate student athletes assessing the amount of daily food intake based on their body weight, an objective indicator. (Lines 401-403)

From line 190-191, please clarify which values are which. For example, potentially revise the sentence from “The total score of body image and that of eating behavior in para-athletes and collegiate student athletes were 30.5 vs. 33.8 and 25.8 vs. 29.0, respectively.” to “The total body image and eating behavior scores were 30.5 and 25.8, respectively, in para-athletes and 33.8 and 29.0, respectively, in collegiate athletes.”

In accordance with Reviewer 3’s suggestion, we revised the sentence as follows:

The total body image and eating behavior scores were 30.8 and 26.0, respectively, in para-athletes and 33.8 and 29.0, respectively, in collegiate student athletes. (Lines 240-241)

In Table 5, please modify the spacing so it is clearer which items go with which responses.

              In accordance with Reviewer 3’s comment, we modified the spacing in Table 5.

In all tables that report p-values, please include a note describing which statistical tests were used for easy reference.

In accordance with Reviewer 3’s comment, we added statistical methods to the footnotes of Table 4 and 5 as follows:

The scores of total and each section were compared by Mann-Whitney U-test. (Lines 270-272)

The differences in answers to each question about dietary practices between the groups were examined by chi-squared test. (Lines 283-284)

On line 325, please explain what is meant by lifestyle-related disease and/or provide a couple of examples.

Lifestyle-related diseases refers to diseases such as hypertension and diabetes. People with spinal cord injury have higher prevalence rate of these diseases compared to general population (Imai K et al. J Clin Epidemiol, 1996).

According to Reviewer 3’ suggestion, we revised the sentence as follows:

In the present study, most para-athletes were unaware of the types of fats they should reduce in their diet (Q21 General Nutrition Knowledge Question), despite their attention to weight gain and lifestyle-related diseases such as hypertension and diabetes because of high susceptibility of people with spinal cord injury to those diseases [63]. (Lines 368-372)

On line 351-352, please explain why para-athletes do not have many opportunities to measure their body weight.

Regarding the point Reviewer 3 pointed out, it is difficult for people with lower-limb disability keeping a standing position on the scale. Moreover, wheelchair scales are generally only available in specific facilities such as hospitals and rehabilitation centers. These are why para-athletes do not have many opportunities to measure their body weight.

We added the above content to the revised manuscript. (Lines 405-407)

The sentence from line 368-369 is somewhat confusing because the participants responded that they were either not consuming enough fluid or they were unsure if they were consuming enough. Therefore, it doesn’t seem completely accurate to say that the majority of para-athletes could not meet their fluid requirements despite knowing what they are unless you refer to just the group that answered ‘not adequate’.

We thank Reviewer 3 for helpful comment. As Reviewer 3 pointed out, the description in the original version was not accurate.

We revised the sentence as follows:

 These results suggest that para-athletes cannot meet fluid requirements even if they know the amount of fluid needed to consume, or they are not even aware of how much fluid to take. (Lines 423-425)

Finally, the paper could use some editing to ensure proper English throughout, which will increase its readability.

We corrected the text throughout so that it is appropriate English.

Reviewer 4 Report

No comments.

Author Response

We greatly appreciate Reviewer 4 for kind peer review.

Round 2

Reviewer 1 Report

Thanks for your efforts. All my comments were addressed by the authors.

Author Response

We greatly appreciate Reviewer 1 for kind peer review and comments again.

Reviewer 2 Report

[Major]
1. The authors have revised their manuscript to add the reason for including college athlete students, but I cannot read the reason to investigate college athletes as well. First of all, is there any academic significance in comparing the nutritional knowledge of ordinary para-athletes with an unknown level of competition and athlete students who belongs to highly educated university with a department of nutrition? Is it correct to mention that “para-athletes” have low nutritional knowledge without considering characteristics such as educational background? Is it correct to mention that “para-athletes” wanted more about general nutrition questions without considering characteristics such as age? I feel that the background argument needs to be revamped.
2. Regarding the Nutrition Knowledge Questionnaire as the main indicator, the authors responded “The final questionnaires were completed after many stages of verification regarding accuracy, construct validity, internal reliability, and reproducibility”. However, this refers to the questionnaire used as a reference, not the questionnaire for this study. I believe that if you use for different subject groups, extract items, and modify questions (e.g., Q29), a new examination of the validity and reliability of the questionnaire is necessary.
3. Regarding the eligibility criteria, the authors responded “The participants who answered their major types of disabilities are "Other" or "Unanswered" remained to be included as in the original version because it was confirmed by the staff of the Osaka Para-Sports Association that they were physically disabled.” Details of disability are special care-required personal Information. Is not there an ethical problem in obtaining information from a third party that the participant has chosen "Unanswered"?

[Minor]
L60–65: Athlete students are suddenly mentioned as an example after the topic about para-athletes, which is very disconcerting.
L68–69: Is there no source for the text?
L121–123: How many participants were excluded from the study?
L137: If it is limited to physical, would it be "Major types of physical disabilities"?
L153: Since it is related to the validity of the questionnaire, I think you should describe specifically what "partly" means.
L171: Since the number of participants has changed, will the coefficient alpha also change?
L173: I think the coefficient alpha of the Nutrition Knowledge Questionnaire should also be shown.
L253: It would be better to show the standard deviation as well.
L283: It would be better to show the p-value as a real number.
L478: I think you should discuss not only the limitations of the questionnaire's characteristics, but also the problems with the modified contents.

Author Response

We greatly appreciate Reviewer 2 for kind and detailed review. We have made corrections to each of the issues. Our responses are as follows:

The authors have revised their manuscript to add the reason for including college athlete students, but I cannot read the reason to investigate college athletes as well. First of all, is there any academic significance in comparing the nutritional knowledge of ordinary para-athletes with an unknown level of competition and athlete students who belongs to highly educated university with a department of nutrition? Is it correct to mention that “para-athletes” have low nutritional knowledge without considering characteristics such as educational background?

Previous studies on nutrition knowledge of collegiate student athletes and para-athletes show these groups have low nutrition knowledge scores, but there are no studies compared these two groups.

It is an interesting point to consider any academic significance in our two groups. However, this is a limitation of our study that we do not know the education levels of our para-athletes. We mentioned this in the limitation of the study (Lines 481-484)

Less than 10 % of collegiate student athletes were nutrition major in this study (4 students out of 45 students). We have checked the answers of collegiate student athletes based on their majors at the university and found no differences in their answers of nutrition knowledge, except the answers to few questions. Overall, these students score did not affect the nutrition knowledge scores of collegiate student athletes.

Is it correct to mention that “para-athletes” wanted more about general nutrition questions without considering characteristics such as age? I feel that the background argument needs to be revamped.

This is a great point whether or not we are able to generalize the responses of nutrition knowledge and nutrition behavior to every athlete with or without disability, age differences, and other characteristics of these two groups. However, our interests are not only comparing the results of these two groups, but also reporting the needs and demands of qualified dietitians for these totally different athlete groups are the same. We have mentioned this in Introduction as follows:

In that sense, general collegiate student athletes and para-athletes groups have the same needs and demand of having sports dietitians beyond having their physical conditions or disabilities, age differences, and education levels. (Lines 60-63)….Evidence suggests there is needs of sports dietitians for competitive athletes. In other words, this is a huge opportunity for dietitians who are interested in working with athletes in various programs at schools or institutions at professional settings. In the United States, the number of sports dietitians in the collegiate setting has grown exponentially, and the efficacy of sports dietitians for collegiate student athletes and para-athletes have been reported [23-25]. (Lines 67-72)

Regarding the Nutrition Knowledge Questionnaire as the main indicator, the authors responded “The final questionnaires were completed after many stages of verification regarding accuracy, construct validity, internal reliability, and reproducibility”. However, this refers to the questionnaire used as a reference, not the questionnaire for this study. I believe that if you use for different subject groups, extract items, and modify questions (e.g., Q29), a new examination of the validity and reliability of the questionnaire is necessary.

L173: I think the coefficient alpha of the Nutrition Knowledge Questionnaire should also be shown.

L478: I think you should discuss not only the limitations of the questionnaire's characteristics, but also the problems with the modified contents.

We developed current nutrition knowledge questionnaire by translating the existing nutrition knowledge questionnaires in Japanese, then, selected questionnaires that could fit in typical Japanese dietary practices and adjusted some wording without changing contents. In this study we didn’t examine the validity of current questionnaire. As Reviewer 2 pointed out, we should have verified the reliability of current questionnaire as to whether nutritional knowledge in our participants could be evaluated. When we actually calculated the ’Cronbach's alpha of current nutrition knowledge questionnaire according to Reviewer 2’s suggestion, the coefficients of general nutrition and sport nutrition sections were 0.863 and 0.504, respectively, which indicated that internal consistency of sport nutrition questionnaire was moderate. Therefore, in the further study, the questionnaire needs to be improved to be highly reliable to evaluate nutrition knowledge.

We mentioned the issue in the Limitation section in the revised version. (Lines 467-481)

We added sentence about the Cronbach’s alpha of the nutrition knowledge questionnaire used in the current study on the Nutrition Knowledge Questionnaire section as follows:

The Cronbach’s alpha of this general nutrition and sport nutrition sections were 0.863 and 0.504, respectively. This indicates that internal consistency of sport nutrition questionnaire was moderate and need to be caution to interpretate the results. (Lines 188-191)

Regarding the eligibility criteria, the authors responded “The participants who answered their major types of disabilities are "Other" or "Unanswered" remained to be included as in the original version because it was confirmed by the staff of the Osaka Para-Sports Association that they were physically disabled.” Details of disability are special care-required personal Information. Is not there an ethical problem in obtaining information from a third party that the participant has chosen "Unanswered"?

L121–123: How many participants were excluded from the study?

Regarding the point Reviewer 2 pointed out, the staff of the Osaka Para-Sports Association called for participations from teams that is composed of people with physical disability and no other disabilities, such as visually impaired, hearing impaired, and intellectual disabilities or internal diseases apparently from the characteristics of the competitions (examples of teams wheelchair table tennis, sitting volleyball, etc.) among the teams engaged in activities such as regular practice at Fine Plaza Osaka. In this way, it has been confirmed that para-athletes have physical disabilities and are not accompanied by other disabilities. Therefore, we used all the responses obtained for the analysis and no responses were excluded. As Reviewer 2 pointed out, it is an important ethical point and a sensitive issue of para-athletes,  the detailed  disability, which  is personal information requires careful handling. The staff did not obtain information about disability details from individual para-athletes, and we and the staff could not identify the individual from the response content. Thank you  to Reviewer, we realized we needed to add this important point in the section of Participants. The above-mentioned participants selection method also has been approved by the Institutional Review Board. Therefore, we believe that this did not conflict with ethical issues. The previous version didn't fully explain this, so we revised 2.1. Participants section as follows:

We have recruited the study participants from about 200 athletes with physical disabilities and no other disabilities (such as visually impaired, hearing impaired, and intellectual disabilities or internal diseases) who are engaged in activities such as regular practice at the Osaka Prefectural Exchange Promotion Center for Persons with Disabilities, also known as Fine Plaza Osaka (Sakai City, Osaka, Japan, http://www.fineplaza.jp/) through the Osaka Para Sports Association (Sakai City, Osaka, Japan, http://www.osad.jp/ ) staff. The staff called for study participations from teams that is clearly composed of physically disabled people from the characteristics of the competitions (examples of team; wheelchair table tennis, sitting volleyball, etc.) via face-to-face or using leaflets and recruited para-athletes for the online survey. All the responses obtained were used for the analysis. Neither we nor the staff could identify individuals who responded and answered the questionnaire. (Lines 113-124)

L60–65: Athlete students are suddenly mentioned as an example after the topic about para-athletes, which is very disconcerting.

Good point.  We have revised the Introduction to explain why we studied these two groups as follow:

However, not all athletes (both collegiate student athletes and para-athletes) at competitive levels, can receive such every sort of support by full-time sports dietitians [19, 20] because their university or institution are not able to afford hiring them. In that sense, general collegiate student athletes and para-athletes groups have the same needs and demand of having sports dietitians beyond their physical conditions or disabilities, age differences, and education levels. (Lines 58-63)

L68–69: Is there no source for the text?

In fact, our study is the first study to examine two groups i.e., collegiate students and para-athletes. But since comparison of these two groups are not our main interests, thus, we have removed the text.  Also, we have added the Reference to the following sentence:

There actually are very few reports available on dietary intake patterns of para-athletes [11]. (Lines 74-75)

L137: If it is limited to physical, would it be "Major types of physical disabilities"?

According to Reviewer 2’s comment, we have corrected the words “Major types of disabilities” in line 137 to “Major types of physical disabilities”. (Line 138)

L153: Since it is related to the validity of the questionnaire, I think you should describe specifically what "partly" means.

We used 6 out of 33 items regarding eating for health (eating perception), eating proper amounts and balanced meals, measuring their body weight, satiety responsiveness, and eating favorite and non-favorite foods in previously reported questionnaire (Takayama N, et al. Kenkouigaku 21(1): 28, 2012) as a reference when developing the current questionnaire.

We added the above content to the revised manuscript. (Lines 154-158)

L171: Since the number of participants has changed, will the coefficient alpha also change?

According to Reviewer 2’s comment, we re-calculated the coefficient alpha coefficient alpha, and revised as follows:

The Cronbach’s alpha of this questionnaire was 0.7140. (Line 174)

L253: It would be better to show the standard deviation as well.

In accordance with Reviewer 2’s suggestion, we revised the sentence as follows:

The total body image and eating behavior scores were 30.8 ± 7.8 and 26.0 ± 7.4, respectively, in para-athletes and 33.8 ± 9.2 and 29.0 ± 5.1, respectively, in collegiate student athletes. (Lines 255-257)

L283: It would be better to show the p-value as a real number.

In accordance with Reviewer 2’s suggestion, we showed the p-values in Table 4.

Furthermore, we deleted asterisks from Table 4, and the following sentence from the footnote of Table 4.

* p < 0.05 and ** p < 0.01 compared with the score obtained from collegiate student athletes. (Lines 286)